# A comparative analysis of factors associated with modern contraceptive use among youth in Northern and Southern Nigeria: A cross-sectional population-based survey (2011–2021)

Olaniyi Felix Sanni[ID]*, Abike Elizabeth Sanni, Collins Isioma Onyeagwaibe, Tukwasi Ahamuefula[ID], Oluwadare Peter Akeju

Research and Development Department, Fescos of Data Solutions, Ota, Ogun State, Nigeria

* fescosofanalysis@gmail.com

## Abstract

Contraceptive use among youth in Nigeria is crucial for improving reproductive health. Despite global advancements, barriers related to sociodemographic, cultural, and religious factors hinder access to family planning services. This study investigates a comparative analysis of factors associated with modern contraceptive use (MCU) among youth in Northern and Southern Nigeria. Data were sourced from Nigeria's Multiple Indicator Cluster Surveys (MICS) for 2011–2021. A retrospective cross-sectional analysis was performed on adolescents and young women aged 15–24. Descriptive statistics assessed contraceptive prevalence. Logistic regression models evaluated associations between contraceptive use and sociodemographic factors such as age, marital status, education, wealth, and region. Southern Nigeria had higher education levels (85.7% with secondary or higher education) and MCU rates (12.6%), compared to Northern Nigeria (50.7% with secondary or higher education; MCU rate of 3.9%). Wealth was a significant factor in the North (AOR = 2.233, p < 0.001), while education emerged as a strong predictor of MCU in both regions. Religious influence was more pronounced in the South, where Christians were significantly more likely to use contraceptives (AOR = 2.670, p < 0.001).The study highlights the need for region-specific interventions to address the disparities in MCU. In Northern Nigeria, targeted efforts should focus on improving education and socioeconomic conditions while addressing cultural and religious barriers to contraceptive use. Continued support for education and family planning services is essential to maintain and enhance MCU rates in Southern Nigeria. Policymakers should prioritize these tailored approaches to promote equitable access to contraceptives and improve reproductive health outcomes across Nigeria.

**Data availability statement:** All relevant data are within the paper and its Supporting Information files.

**Funding:** The author(s) received no specific funding for this work.

**Competing interests:** The authors have declared that no competing interests exist.

## Author summary

This study examines the factors influencing modern contraceptive use (MCU) among youth in Northern and Southern Nigeria from 2011 to 2021. Using data from the Multiple Indicator Cluster Surveys (MICS), we conducted a comparative analysis to identify key determinants of MCU, including age, education, wealth, marital status, and religious influence. Findings reveal significant regional disparities, with Southern Nigeria exhibiting a higher MCU rate (12.6%) compared to Northern Nigeria (3.9%). Education was a strong predictor in both regions, while wealth played a more significant role in the North. Religious and cultural barriers were more pronounced in the North, limiting contraceptive uptake. Trends in MCU fluctuated over the decade, reflecting inconsistencies in access and acceptance. This study underscores the need for region-specific interventions to address barriers to contraceptive use. In Northern Nigeria, efforts should focus on improving education, expanding access to family planning services, and engaging religious and community leaders. Meanwhile, sustaining educational and healthcare investments is crucial in the South. Policymakers must prioritize youth-friendly reproductive health programs and awareness campaigns to bridge the contraceptive gap and promote equitable access for Nigerian youth.

## Introduction

Based on the 2020 World Youth Report, the expected number of individuals aged 15–24 years was 1.2 billion, representing 16% of the world population [1,2]. In 2021, the United Nations anticipated that the population of young people will have increased by 7% to reach about 1.3 billion by 2030, the target year for the Sustainable Development Goals (SDGs) [2]. Adolescents and young people in humanitarian crises face substantial challenges due to the disruption of their support system, making them more vulnerable to sexual coercion, exploitation, violence, early marriage, and reliance on harmful coping strategies [3,4]. These factors can contribute to an increase in unintended pregnancies and a higher demand for safe abortion services, among other negative consequences [3].

Globally, the use of contraceptives has risen steadily over the years, with the global contraceptive prevalence rate (CPR) for modern methods reaching 49% in 2020 [4]. In 2021, among 15–19-year-olds, modern contraceptive use in Sub-Saharan Africa stood at about 10 percent, compared with approximately 20 percent in Australia and New Zealand and roughly 30 percent in Europe and Northern America. Similarly, among 20–24-year-olds, the rate in Sub-Saharan Africa was around 25 percent, while it reached nearly 50 percent in Latin America and the Caribbean and Europe and Northern America, and approached 60 percent in Australia and New Zealand [5]. However, some countries within SSA such like Kenya, South Africa, and Ethiopia have made substantial progress in increasing contraceptive uptake among young women through targeted policies and programs [6,7]. In contrast, Nigeria continues to face major challenges in improving contraceptive uptake [8].

The 2018 Nigeria Demographic and Health Survey (NDHS) revealed that only 12% of sexually active young women aged 15–24 use modern contraceptives [9]. This finding is in contrast to countries like Ghana and Senegal, where concerted efforts have led to higher contraceptive use among young women [10]. A significant portion of Nigeria's population comprises adolescents and young individuals. It is estimated that by 2025, the country will have around 57 million young people. Studies show that approximately 28% of Nigerian adolescents participate in sexual activities, with their initiation into sexual activity occurring between the ages of 10 and 15 [11]. Nigeria holds the second position in terms of maternal mortality, and illegal and unsafe abortions contribute to a significant portion, approximately 20% to 40%, of the country's annual maternal death toll [12]. Providing adolescents and young women with access to modern contraceptives and comprehensive sexual and reproductive health services has been demonstrated to yield positive health outcomes [12].

The lack of contraceptive use among young women in Nigeria exacerbates the high rate of unsafe abortions, which remains a major public health issue [13]. Estimates suggest over 1.25 million abortions occur annually in Nigeria, with many being dangerous, leading to significant morbidity and mortality, especially among young women [13]. Providing effective modern contraceptive access to adolescents and young women can alleviate the public health burden of unsafe abortions and support reproductive autonomy [14].

Access to modern contraceptives is further hindered by inadequate healthcare infrastructure, particularly in rural areas, and high costs associated with contraceptive methods [15]. Many young women are also discouraged by stigma, fear of judgment, and misconceptions about contraceptive side effects [15]. There is a clear need to develop youth-friendly reproductive health services to improve access and usage among adolescents [16]. Despite the importance of contraceptive use in addressing reproductive health challenges, there is a need for research on the long-term trends and determinants of modern contraceptive use (MCU) among young women in Nigeria. While some studies have examined regional disparities, there is limited comprehensive data across Nigeria's geopolitical zones, and few studies have explored how access and barriers evolve [17]. This study aims to address this gap by conducting a comparative cross-sectional analysis of factors associated with MCU among adolescents and young women in Northern and Southern Nigeria from 2011 to 2021. Understanding the trends and factors influencing MCU over the past decade will inform targeted interventions to improve access to contraceptives for adolescents and young women in Nigeria.

## Methodology

### Ethics statement

This study utilised publicly available secondary data from the MICS, which received ethical clearance from Nigerian authorities. MICS obtained informed consent from participants, ensuring confidentiality throughout. Authorisation to access the MICS dataset was granted via an online application outlining the study's objectives. Detailed personal information was excluded from the datasets to protect respondent anonymity.

### Study design

This study adopted a retrospective cross-sectional, population-based survey design to investigate contraceptive utilization among adolescents and young women in Nigeria between 2011 and 2021. The design was chosen because it enables the analysis of historical data over a defined period, making it suitable for identifying trends and patterns in contraceptive use. Secondary data from the Multiple Indicator Cluster Surveys (MICS) were utilized, as these datasets offer a nationally representative sample of households, including detailed information on reproductive health indicators. The availability of high-quality, pre-collected data facilitated efficient and cost-effective research, eliminating the need for primary data collection. The study focused on responses from adolescents and young women aged 15–24 who had exclusively used modern contraceptive methods. This design allowed for the exploration of sociodemographic factors influencing contraceptive use

and provided insights into changes and determinants over the decade. The retrospective approach was particularly valuable for understanding long-term trends and evaluating policy or programmatic impacts on reproductive health behaviour in Nigeria. Respondents with incomplete information on key variables, particularly contraceptive use, were excluded to ensure reliable findings. A complete-case analysis approach was adopted, given the minimal proportion of missing data, which was unlikely to introduce bias. This ensured that only records with complete data were included, maintaining data integrity and representativeness. This study utilized nationally representative data from the Multiple Indicator Cluster Surveys (MICS) spanning 2011–2021. Because the surveys employ a rigorous multistage cluster sampling design with pre-defined sample sizes, an independent sample size calculation was not performed. Instead, all women aged 15–24 years who met the inclusion criteria were included in the analysis. This method is consistent with other studies using large-scale demographic surveys, where the sampling framework itself establishes the sample size [18–20].

## Study area and population

This study centres on Nigeria, the most densely populated nation in Africa, renowned for its exceptional cultural, ethnic, and socioeconomic heterogeneity. Nigeria's geographical division consists of six geopolitical zones: North-East, North-West, North-Central, South-West, South-East, and South-South. These zones encompass 36 states and the Federal Capital Territory (Abuja). The study employs Comprehensive Multiple Indicator Cluster Survey (MICS) statistics from all six zones, spanning the timeframe from 2011 to 2021. The research sample comprises adolescent and young women between the ages of 15 and 24, selected from both Northern and Southern geographic regions for examination using quantitative methods. A total of 22,229 participants were selected from the North-Central (7507), North-East (6524), and North-West (8198) in Northern Nigeria. In Southern Nigeria, an additional 12,951 individuals were selected from the South-East (4193), South-South (4672), and South-West (4086).

## Sampling technique

The MICS surveys employ a multistage cluster sampling design to obtain nationally representative household samples across Nigeria's diverse regions. The sampling process begins with the selection of enumeration areas (EAs) defined by the national census, which serve as primary sampling units (PSUs). These EAs are chosen using probability proportional to size (PPS) based on the population within each area. To ensure adequate representation, the country is stratified by region and by rural and urban areas. Within each selected PSU, a comprehensive listing of households is conducted, followed by the systematic random selection of households for participation. In each selected household, eligible women of reproductive age, including adolescents and young women, are identified. Data collection is carried out from consenting participants using the standardized MICS questionnaire.

## Study instrument

The MICS employed a standardized, well-structured questionnaire developed by UNICEF. This instrument is globally recognized and tailored for each participating country, including Nigeria, to suit local contexts. The questionnaire was tested and validated by the national MICS implementation team to ensure cultural appropriateness and data reliability. It comprised both open-ended and closed-ended questions to capture comprehensive information from consenting reproductive adolescents and young women in Nigeria. As this study utilized secondary data from MICS, no further modifications to the questionnaire were made by the researchers.

## Data source

The data were sourced from the MICS conducted between 2011 and 2021, a standardised survey program developed by UNICEF to deliver internationally comparable, high-quality data. The surveys cover key indicators like health, education,

child protection, and reproductive health. The dataset provides individual-level information on respondents' sociodemographic characteristics and their use of modern contraceptives.

### Dependent variable

The primary outcome variable was MCU, which was measured based on self-reported use of any modern contraceptive methods.

### Independent variables

Key independent variables included:

Age: Categorised into two groups (15–19 and 20–24).

Marital Status: Classified as single, married, or divorced.

Geopolitical Region: The six geopolitical zones (North-Central, North-East, North-West, South-East, South-South, and South-West).

Educational Attainment: Categorised into no formal education, primary education, and secondary education or higher.

Wealth Status: Categorised into poor, middle class, and rich based on household income.

Religion: Categorised as Christianity, Islam, and other faiths.

Parity: Whether the respondent had given birth or not.

### Data analysis

All statistical analyses were conducted using SPSS version 28.0, employing both descriptive and inferential methods. The distributions of categorical variables, including age category, marital status, regional area, education level, wealth index/status, religion, and reproductive history, were summarized using frequencies and percentages. For instance, age was categorized as "15–19" and "20–24," while marital status was grouped as "Single," "Divorced," or "Married." Similar frequency distributions were provided for the remaining variables. The outcome variable, "currently using a method to avoid pregnancy" (Yes/No), was also summarized as percentages. To assess the relationship between each independent variable and the outcome, univariate logistic regression analyses were performed separately for Northern and Southern Nigeria. This analysis generated crude odds ratios (COR) with corresponding 95% confidence intervals (CI) and p-values. Variables that showed significant associations ($p < 0.05$) in the univariate models were further analyzed using multivariate logistic regression to control for potential confounders. Adjusted odds ratios (AOR) with 95% CIs and p-values were reported, with statistical significance set at $p < 0.05$.

## Results

### Demographic characteristics of the study participants

Table 1 presents the sociodemographic characteristics of participants from Northern (N = 22,229) and Southern Nigeria (N = 12,951). The majority were aged 15–19 years (North: 55.6%, South: 57.5%), with the remainder aged 20–24. Marital status differed significantly, with 59.0% of Northern participants being single and 39.1% married, compared to 82.5% single and 15.5% married in the South. Education levels were higher in the South (85.7% with secondary or higher education) than in the North (50.7%). Additionally, 36.3% of Northerners had no formal education, compared to 3.9% in the South. Wealth disparities were notable, with 52.5% of Northern participants classified as poor compared to 20.5% in the South, while 54.4% of Southern participants were wealthy compared to 26.1% in the North. In the North, Christianity was 27.8%, Islam (17.3%) and other religions (12.3%) compared to Christianity 27.3%), Islam (15.6%) and different religions (20.7%) in the South. Northern participants (64.2%) had never given birth, compared to 79.8% in the South. Regarding reproductive health, 5.5% of Northern participants used contraception, compared to 17.9% in the South.

**Table 1.** Sociodemographic characteristics of the study participants.

| Variables | Northern Nigeria (N = 22,229) | | Southern Nigeria (N = 12,951) | |
|---|---|---|---|---|
| | Frequency | Percentage | Frequency | Percentage |
| Age Category | | | | |
| 15-19 | 12369 | 55.6 | 7452 | 57.5 |
| 20-24 | 9860 | 44.4 | 5499 | 42.5 |
| Marital Status | | | | |
| Single | 13107 | 59.0 | 10690 | 82.5 |
| Divorced | 401 | 1.8 | 248 | 1.9 |
| Married | 8689 | 39.1 | 2011 | 15.5 |
| Regional Area | | | | |
| North-Central | 7507 | 33.8 | – | – |
| North-East | 6524 | 29.3 | – | – |
| North-West | 8198 | 36.9 | – | – |
| South-East | – | – | 4193 | 32.4 |
| South-South | – | – | 4672 | 36.1 |
| South-West | – | – | 4086 | 31.5 |
| Education Level | | | | |
| None | 8065 | 36.3 | 501 | 3.9 |
| Primary | 2889 | 13.0 | 1354 | 10.5 |
| Secondary & above | 11271 | 50.7 | 11095 | 85.7 |
| Wealth Index/Status | | | | |
| Poor | 11673 | 52.5 | 2661 | 20.5 |
| Middle | 4754 | 21.4 | 3245 | 25.1 |
| Rich | 5802 | 26.1 | 7045 | 54.4 |
| Religion | | | | |
| Christianity | 6177 | 27.8 | 3542 | 27.3 |
| Islam | 3854 | 17.3 | 2020 | 15.6 |
| Others | 2797 | 12.6 | 2679 | 20.7 |
| Ever giving birth | | | | |
| No | 14268 | 64.2 | 10337 | 79.8 |
| Yes | 7943 | 35.7 | 2610 | 20.2 |
| Currently using a method to avoid pregnancy. | | | | |
| No | 21008 | 94.5 | 10638 | 82.1 |
| Yes | 1221 | 5.5 | 2313 | 17.9 |

## MCU among youth in Northern and Southern Nigeria

Fig 1 shows MCU among youth in Northern and Southern Nigeria. In the Northern region, 96.1% of youth do not use modern contraceptives, while only 3.9% do. In contrast, Southern Nigeria has a non-use rate, with 87.4% not using modern contraceptives and 12.6% reported using.

## Trend of MCU use among youth in Nigeria 2011–2021

Fig 2 represents the trend of MCU among youth in Northern and Southern Nigeria from 2011 to 2021. In the Northern region, usage fluctuated, starting at 4.3% in 2011, dropping to 1.2% in 2017, and rising to 4.6% by 2021. In the Southern region, usage decreased from 18.1% in 2011 to 8.1% in 2016 but then increased to 11.9% by 2021.

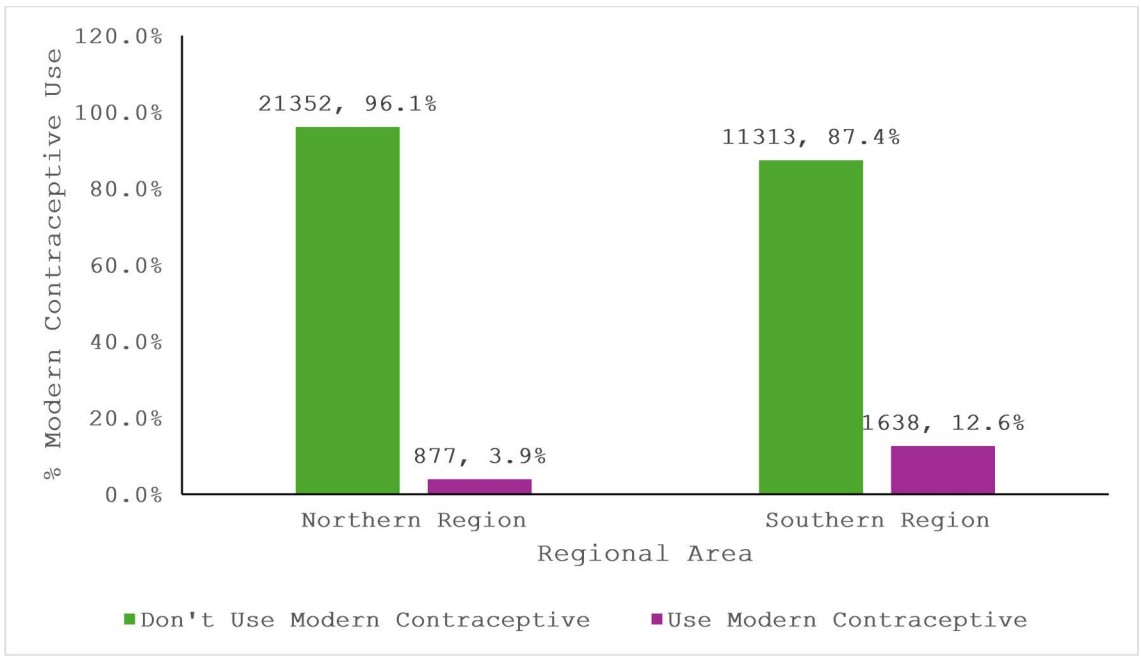

**Fig 1. MCU among youth in Northern and Southern Nigeria.**

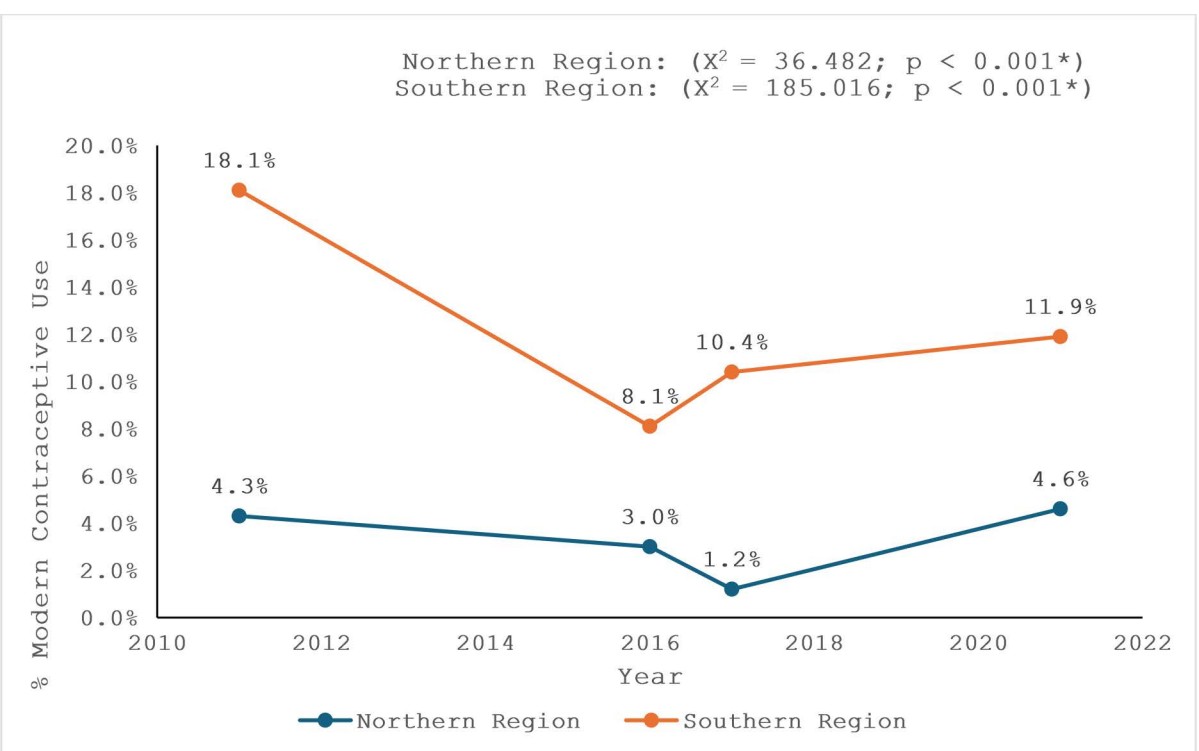

**Fig 2. Trend of MCU among youth in Nigeria 2011 - 2021.**

## Factors associated with MCU among youth in Northern and Southern Nigeria

Table 2 compares factors associated with MCU among youth in Northern and Southern Nigeria using univariate and multivariate analysis. Youth aged 20–24 in both regions were significantly more likely to use modern contraceptives than those aged 15–19. In Northern Nigeria (AOR = 3.321, p < 0.001), while in Southern Nigeria (AOR = 3.153, p < 0.001). In the univariate analysis, being divorced significantly increased the likelihood of MCU in both regions. Married youth in both areas were less likely to use contraceptives in the adjusted models, with AORs of 0.506 (p = 0.001) in the North and 0.559 (p < 0.001) in the South. In Northern Nigeria, the North-Central region had significantly higher contraceptive use (AOR = 3.352, p < 0.001) compared to the North-West. In Southern Nigeria, the South-East and South-South regions also showed significantly higher odds of contraceptive use compared to the South-West (AOR = 1.350, p < 0.001 and 1.242, p = 0.014, respectively). Higher education levels were associated with increased contraceptive use in both regions. In

**Table 2. Factors associated with MCU among youth in Northern and Southern Nigeria.**

| Variables | NORTHERN NIGERIA | | | | SOUTHERN NIGERIA | | | |
|---|---|---|---|---|---|---|---|---|
| | UNIVARIATE | | MULTIVARIATE | | UNIVARIATE | | MULTIVARIATE | |
| | COR (95% CI) | P-value | AOR (95% CI) | P-value | COR (95% CI) | P-value | AOR (95% CI) | P-value |
| **Age Category (15–19)** | | | | | | | | |
| 15-19 | Ref. | – | – | – | Ref. | – | – | – |
| 20-24 | 3.775 [3.238-4.401] | <0.001* | 3.321 [2.622-4.207] | <0.001* | 2.992 [2.683-3.336] | <0.001* | 3.153 [2.728-3.644] | <0.001* |
| **Marital Status** | | | | | | | | |
| Single | Ref. | – | – | – | Ref. | – | – | – |
| Divorced | 2.544 [1.751-3.697] | <0.001* | 1.104 [0.591-2.061] | 0.756 | 2.342 [1.744-3.145] | <0.001* | 1.306 [0.838-2.035] | 0.238 |
| Married | 1.464 [1.276-1.680] | <0.001* | 0.506 [0.336-0.763] | 0.001* | 1.113 [0.967-1.281] | 0.135 | 0.559 [0.420-0.743] | <0.001* |
| **Regional Area** | | | | | | | | |
| North-Central | 3.377 [2.843-4.011] | <0.001* | 3.352 [2.560-4.388] | <0.001* | – | – | – | – |
| North-East | 1.177 [0.952-1.457] | 0.133 | 1.444 [1.052-1.983] | 0.023* | – | – | – | – |
| North-West | Ref. | – | – | – | – | – | – | – |
| South-East | – | – | – | – | 1.704 [1.485-1.956] | <0.001* | 1.350 [1.122-1.625] | 0.001* |
| South-South | – | – | – | – | 1.702 [1.487-1.948] | <0.001* | 1.242 [1.044-1.478] | 0.014* |
| South-West | – | – | – | – | Ref. | – | – | – |
| **Education Level** | | | | | | | | |
| None | Ref. | – | – | – | Ref. | – | – | – |
| Primary | 1.361 [1.068-1.735] | 0.013* | 1.536 [1.046-2.254] | 0.029* | 1.011 [0.714-1.431] | 0.951 | 1.174 [0.591-2.333] | 0.647 |
| Secondary & above | 2.022 [1.720-2.377] | <0.001* | 1.934 [1.394-2.684] | <0.001* | 1.429 [1.056-1.934] | 0.021* | 1.937 [1.020-3.678] | 0.043* |
| **Wealth Index/Status** | | | | | | | | |
| Poor | Ref. | – | – | – | Ref. | – | – | – |
| Middle | 1.743 [1.457-2.085] | <0.001* | 1.463 [1.112-1.925] | 0.007* | 1.060 [0.908-1.238] | 0.462 | 1.000 [0.808-1.238] | 0.999 |
| Rich | 2.504 [2.143-2.927] | <0.001* | 2.233 [1.740-2.866] | <0.001* | 1.047 [0.915-1.199] | 0.504 | 1.155 [0.950-1.404] | 0.148 |
| **Religion** | | | | | | | | |
| Christianity | 0.638 [0.511-0.797] | <0.001* | 1.022 [0.806-1.295] | 0.857 | 2.622 [2.229-3.085] | <0.001* | 2.670 [2.236-3.188] | <0.001* |
| Islam | 0.577 [0.447-0.744] | <0.001* | 1.136 [0.861-1.499] | 0.367 | 1.226 [1.002-1.501] | 0.048* | 1.171 [0.930-1.474] | 0.180 |
| Others | Ref. | – | – | – | Ref. | – | – | – |
| **Ever giving birth** | | | | | | | | |
| No | Ref. | – | – | – | Ref. | – | – | – |
| Yes | 2.022 [1.766-2.316] | <0.001* | 2.174 [1.459-3.239] | <0.001* | 1.310 [1.159-1.480] | <0.001* | 1.025 [0.792-1.328] | 0.849 |

Source: Field Survey Conducted in Nigeria *Significant at p < 0.05, COR = Crude Odds Ratio; AOR = Adjusted Odds Ratio; CI = Confidence Interval

Northern Nigeria, secondary education or above had an AOR of 1.934 (p<0.001), and in Southern Nigeria, the AOR was 1.937 (p=0.043). In Northern Nigeria, youth from wealthier households had a higher likelihood of using contraceptives (AOR=2.233, p<0.001), whereas wealth did not significantly impact contraceptive use in the South. Religion played a significant role in Southern Nigeria, where Christians were much more likely to use contraceptives (AOR=2.670, p<0.001), while Islam showed no significant association. In Northern Nigeria, there was no significant difference based on religion in the multivariate model. Participants who have given birth increased the odds of contraceptive use (AOR=2.174, p<0.001) in the North, but this association was not significant in Southern Nigeria.

**Modern contraceptive methods use among youth in Northern and Southern Nigeria**

Fig 3 illustrates the use of modern contraceptive methods among youth in Northern and Southern Nigeria. In both regions, sterilisation and IUD use are minimal at 0.1%. In Northern Nigeria, 2.5% of youth use injectables, implants, and pills, while Southern Nigeria has a higher usage rate of 4.3%. Condom and diaphragm use is significantly higher in the South at 9.7%, compared to 1.7% in the North.

**Reasons for not using contraceptive among youth in Northern and Southern Nigeria**

Fig 4 outlines the reasons for not using contraceptives among youth in Northern and Southern Nigeria. A small percentage of participants in both regions were trying to get pregnant (0.3% in the North and 0.4% in the South). Northern youth more frequently cited no or infrequent sexual activity (7.4% compared to 5.7% in the South) and breastfeeding (4.1) compared to the South (1.1%) as reasons for non-use. Menstrual issues were reported at a higher rate in the North (1.7%) compared to 0.7% in the South. Medical conditions such as hysterectomy and amenorrhea were reported by 2.3% in the North compared to 1.0% in the South as the reason for not using contraceptives among the youth.

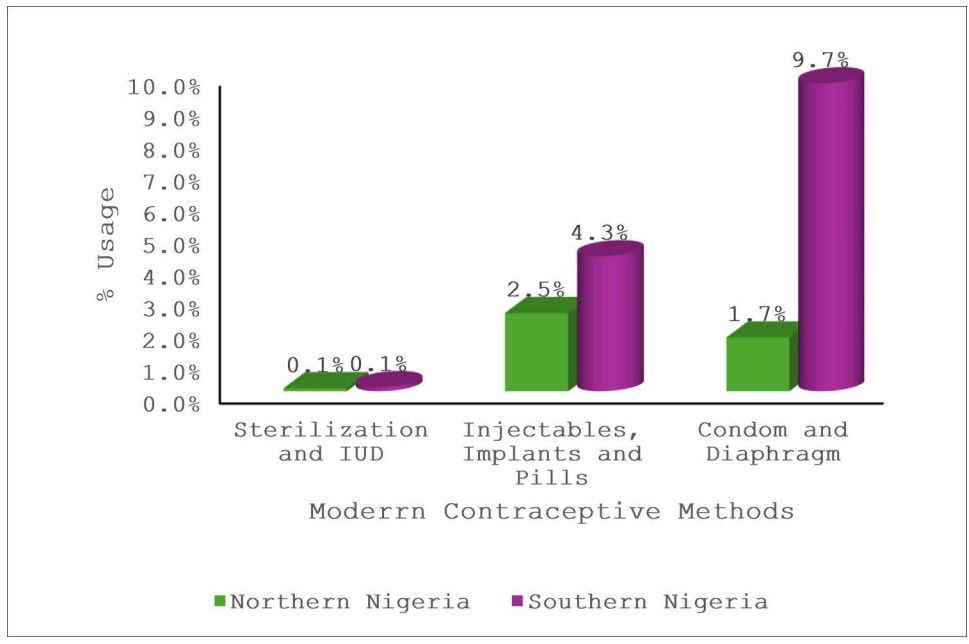

**Fig 3. Modern contraceptive methods use among youth in Northern and Southern Nigeria.**

**Fig 4. Reasons for not using contraceptive among Northern and Southern YOUTH.**

## Discussion

### Key findings summary

This study examined the factors influencing modern contraceptive use (MCU) among youth in Northern and Southern Nigeria between 2011 and 2021. Findings revealed stark regional disparities, with Southern Nigeria reporting a significantly higher MCU rate (12.6%) compared to Northern Nigeria (3.9%). Factors such as age, education, wealth, and marital status were significant predictors of MCU in both regions, though their impact varied. While wealth played a critical role in the North, education was the strongest predictor in both regions. Religious influence was more pronounced in the South, where Christians were more likely to use modern contraceptives. Trends in MCU fluctuated over the study period, with notable declines between 2011 and 2016, followed by a gradual recovery in both regions.

### Demographic characteristics of the study participants

The demographic characteristics of this study reveal significant regional differences, particularly in education, wealth, and marital status. In Southern Nigeria, 85.7% of participants had secondary or higher education, compared to only 50.7% in the North. These findings align with Oluwatobi [21] who found a similar divide in education, noting that education levels were higher in the South, which influenced MCU. In Oluwatobi's study, factors such as education and wealth were significantly associated with MCU in the South. At the same time, socioeconomic challenges were more pronounced in the North, where 52.5% of participants in the current study were classified as poor, compared to just 20.5% in the South.

The MCU in Northern Nigeria (3.9%) is substantially lower than in Southern Nigeria (12.6%). These findings are consistent with Michael *et al.* [22], where Chad exhibited the lowest prevalence of MCU at 4%, similar to Northern Nigeria, and Namibia had the highest prevalence at 60.5%, akin to Southern Nigeria's relatively higher rates. The MCU in the North

aligns with Oluwatobi [21], who also reported a 3.6% prevalence in Northern Nigeria, while Southern Nigeria's 12.6% is closer to the 8.6% MCU reported in the same study. In contrast, the higher rates in the South align with Amoah *et al.* [23], where higher education and access to contraceptives increased utilisation.

### Trend of MCU use among youth in Nigeria 2011–2021

The fluctuating trend in MCU use observed in the Northern region between 2011 and 2021 (from 4.3% to 1.2% and back to 4.6%) indicates inconsistencies in contraceptive access and social acceptance. For instance, the National Family Planning Blueprint (2014–2020) aimed to increase contraceptive prevalence by improving access to modern contraceptives, particularly through task-shifting policies that allowed community health workers to provide contraceptive services [24]. However, persistent sociocultural and religious opposition to contraception in the North may have limited the impact of such policies [25]. Additionally, disruptions caused by security challenges in the region, such as insurgency and internal displacement, may have further restricted access to reproductive health services [26]. In Southern Nigeria, MCU declined from 18.1% in 2011 to 8.1% in 2016 before rising to 11.9% in 2021. This trend could be attributed to multiple factors, including variations in government funding, the implementation of family planning programs, and the expansion of private sector contraceptive distribution [24]. The rise in MCU after 2016 may have been influenced by increased donor support, particularly through the FP2020 initiative, which sought to improve contraceptive access for women and youth [24]. Additionally, media campaigns and digital health initiatives, such as the NURHI (Nigerian Urban Reproductive Health Initiative), may have contributed to improved awareness and uptake of modern contraception in the South [24]. In contrast to Ezire *et al.* [27], who found a 0.47% annual increase in MCU between 2003 and 2011; the current study shows fluctuating trends, with MCU in Northern Nigeria dropping to 1.2% in 2017 before rising to 4.6% in 2021. This inconsistency suggests challenges in maintaining sustained MCU in the North, where factors such as low education, poverty, and religious influences might hinder progress. In Southern Nigeria, MCU fell from 18.1% to 8.1% between 2011 and 2016 before rising to 11.9% by 2021. This trend is mirrored by Manet *et al.* [28], who noted that improved access to contraception in urban areas, such as in Southern Nigeria, can reverse declining trends.

### Factors associated with MCU among youth in Northern and Southern Nigeria

The study highlighted age, education, wealth, and region as significant factors affecting contraceptive use in both Northern and Southern Nigeria. Youth aged 20–24 in both areas were more likely to use modern contraceptives than those aged 15–19, with adjusted odds ratios (AOR) of 3.321 (p < 0.001) in the North and 3.153 (p < 0.001) in the South. This finding aligns with Asiimwe *et al.* [6], who found that younger married women aged 15–24 were less likely to use contraceptives compared to older women, reflecting the importance of age as a determinant. Similarly, Hounton *et al.* [29] noted that younger, unmarried women were more likely to use contraceptives, corroborating the trend observed in Southern Nigeria.

### Educational and economic influences

Education emerged as a strong factor in both regions, with secondary education or higher associated with higher odds of contraceptive use (AOR = 1.934, p < 0.001 in the North; AOR = 1.937, p = 0.043 in the South). This result is in agreement with Oyinlola *et al.* [30], where women with higher education in South-West Nigeria had significantly higher MCU rates (OR=1.35, CI: 1.045-1.754). Wealth was also a significant factor in Northern Nigeria (AOR = 2.233, p < 0.001), whereas in the South, wealth did not significantly impact contraceptive use (AOR = 1.155, p = 0.148). This result differs from Oluwatobi [21], who found wealth to be a significant factor in both regions, suggesting that other socio-cultural dynamics may play a role in Southern Nigeria. Contextually, the weaker association between wealth and MCU in the South may reflect greater accessibility of contraceptive services, which reduces financial barriers to use. Southern Nigeria has a more developed healthcare infrastructure, including private sector involvement in contraceptive distribution, making cost less of a limiting

factor. In contrast, Northern Nigeria has higher poverty levels and limited healthcare access, meaning wealthier individuals may have better opportunities to seek contraceptive services, explaining the stronger association between wealth and MCU in this region [31,32].

### Religious and cultural barriers

Religion played a crucial role in Southern Nigeria, where Christians were significantly more likely to use contraceptives (AOR = 2.670, p < 0.001), while Islam had no significant association. This finding contrasts with Oluwatobi [21], where religion was an important factor in the North, with Islamic beliefs inhibiting contraceptive use. The lack of religious influence in the North in this study may reflect changing dynamics, though Nkenguye *et al.* [33] found that being married or cohabiting (AOR = 5.7) significantly increased contraceptive use, highlighting the complex relationship between religion, marital status, and contraception in different regions. It suggests that while religion is a prominent factor in some areas, its influence can vary greatly depending on local socio-cultural dynamics. Beyond formal religious doctrines, traditional beliefs and gender norms critically shape contraceptive behaviours in Northern Nigeria. Pronatalist cultural expectations (where large families are seen as symbols of prosperity and social status) discourage contraceptive use, particularly among married women who may view limiting childbirth as contrary to familial and societal norms [34]. Moreover, entrenched gender norms often require women to obtain spousal approval before accessing contraceptives, thereby limiting their reproductive autonomy. This dynamic fosters a preference for traditional or natural methods, driven by concerns over side effects and pervasive myths and misconceptions [35].

### Regional difference in MCU

The methods of contraception also showed regional differences. In Southern Nigeria, condom and diaphragm use was significantly higher (9.7%) compared to Northern Nigeria (1.7%), consistent with Ezire *et al.* [27], where 86% of sexually active women in the South South used condoms. In Northern Nigeria, short-acting methods like injectables, implants, and pills were used by 2.5% of youth, compared to 4.3% in the South. Michael *et al.* [22] also reported a higher prevalence of short-term methods among less-educated youth, aligning with the findings for Northern Nigeria. In both regions, sterilisation and IUD use remained minimal at 0.1%, reflecting similar findings from Oluwatobi. [21], where long-term methods accounted for just 3% of all MCU. Conversely, Olela *et al.* [36] noted that in regions like the Democratic Republic of Congo, condom use remained the dominant method, similar to the Southern Nigerian context.

The reasons for not using contraceptives, including low sexual activity and breastfeeding, were more commonly cited in Northern Nigeria. For instance, 7.4% of Northern youth cited infrequent sexual activity, compared to 5.7% in the South. This finding is consistent with Dioubaté *et al.* [28], who found that in Guinea, lack of sexual activity and myths about contraceptives were common reasons for non-use. Similarly, Cudjoe *et al.* [37] found peer influence and misinformation as significant barriers to MCU. The higher incidence of medical reasons such as hysterectomy and amenorrhea in the North (2.3% compared to 1.0% in the South) further indicates a greater need for targeted family planning programs addressing health-related misconceptions. These barriers suggest that family planning programs in Northern Nigeria need to focus on debunking myths and increasing awareness about the benefits of modern contraceptives, as proposed by Amoah *et al.*. [23] and Matemba & Kautsire [38]. Policymakers and public health practitioners must prioritize comprehensive sexual health education, youth-friendly services, and equitable contraceptive access. Sustained funding and collaboration with community and religious leaders are essential to bridging gaps in awareness and acceptance. Investing in these efforts now will empower young people to make informed reproductive choices and improve health outcomes nationwide.

### Limitations

As with most survey-based studies, the reliance on self-reported data introduces the potential for bias. Participants may underreport or overreport contraceptive use due to social desirability bias, particularly in regions where discussing contraception remains culturally sensitive. Additionally, some key variables, such as detailed cultural beliefs and individual

access to healthcare services, were not captured in the MICS dataset. Their omission may have influenced the estimated impact of education, wealth, and other predictors. The study also faced challenges with missing or incomplete data, which could affect the robustness of the statistical analyses. Future research should incorporate qualitative methods and complementary datasets to provide a more comprehensive understanding of the factors shaping contraceptive use among youth in Nigeria.

## Implications for future studies

1. Future research should employ longitudinal designs to examine changes in contraceptive use over time and identify causal relationships.

2. Investigating the effectiveness of social media and mobile-based family planning campaigns could provide insights into new approaches to increasing contraceptive uptake.

3. In-depth qualitative studies could explore cultural and religious narratives influencing contraceptive use, particularly in Northern Nigeria.

4. Cross-country studies could assess how contraceptive trends in Nigeria compare with those in other West African nations, providing broader insights into regional contraceptive behaviours.

## Conclusion

This study highlights significant regional disparities in MCU among Nigerian youth, with Southern Nigeria exhibiting higher uptake than the North. Key determinants include education, wealth, and marital status, while cultural and religious influences remain substantial barriers in the North. Fluctuating MCU trends over the decade suggest inconsistent progress, particularly in the North, where misconceptions and limited access persist. In the North, efforts should focus on expanding education, increasing family planning access, and addressing cultural barriers through community engagement. Policymakers must prioritize youth-friendly reproductive health programs, integrate contraception into broader healthcare services, and strengthen public awareness campaigns to dispel misconceptions.

## Supporting information

**S1 Text. DHS authorization letter contraceptives.**
(PDF)

**S1 Data. Youth contraceptive data.**
(SAV)

## Author contributions

**Conceptualization:** Olaniyi Felix Sanni, Abike Elizabeth Sanni.

**Data curation:** Olaniyi Felix Sanni, Tukwasi Ahamuefula.

**Formal analysis:** Olaniyi Felix Sanni, Tukwasi Ahamuefula, Oluwadare Peter Akeju.

**Funding acquisition:** Olaniyi Felix Sanni.

**Investigation:** Olaniyi Felix Sanni, Abike Elizabeth Sanni, Collins Isioma onyeagwaibe, Tukwasi Ahamuefula.

**Methodology:** Olaniyi Felix Sanni, Abike Elizabeth Sanni, Collins Isioma onyeagwaibe, Tukwasi Ahamuefula, Oluwadare Peter Akeju.

**Project administration:** Olaniyi Felix Sanni, Abike Elizabeth Sanni.

**Resources:** Olaniyi Felix Sanni, Oluwadare Peter Akeju.

**Software:** Olaniyi Felix Sanni, Tukwasi Ahamuefula.

**Supervision:** Olaniyi Felix Sanni, Abike Elizabeth Sanni.

**Validation:** Olaniyi Felix Sanni.

**Visualization:** Olaniyi Felix Sanni, Tukwasi Ahamuefula, Oluwadare Peter Akeju.

**Writing – original draft:** Olaniyi Felix Sanni, Collins Isioma onyeagwaibe, Oluwadare Peter Akeju.

**Writing – review & editing:** Olaniyi Felix Sanni, Abike Elizabeth Sanni, Collins Isioma onyeagwaibe, Tukwasi Ahamuefula.

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
