## [Decision Letter · Decision Letter 0]

3 Feb 2025

PGPH-D-24-02207

A Comparative Analysis of Factors Associated with Modern Contraceptive Use among Youth in Northern and Southern Nigeria: A Cross-sectional Population-based Survey (2011-2021)

Dear Dr. Sanni,

Thank you for submitting your manuscript to PLOS Global Public Health. After careful consideration, we feel that it has merit but does not fully meet PLOS Global Public Health’s publication criteria as it currently stands. Therefore, we invite you to submit a revised version of the manuscript that addresses the points raised during the review process.

Please provide point-to-point response to the reviewer's comments. 

We look forward to receiving your revised manuscript.

Kind regards,

Tanmay Bagade, Ph.D., MS (O&G), MPH, MHM

Academic Editor

Journal Requirements:

1. Please provide an Author Summary. This should appear in your manuscript between the Abstract (if applicable) and the Introduction, and should be 150–200 words long. The aim should be to make your findings accessible to a wide audience that includes both scientists and non-scientists. Sample summaries can be found on our website under Submission Guidelines:

https://journals.plos.org/globalpublichealth/s/submission-guidelines#loc-parts-of-a-submission.

2. Figure 1: please (a) provide a direct link to the base layer of the map (i.e., the country or region border shape) and ensure this is also included in the figure legend; and (b) provide a link to the terms of use / license information for the base layer image or shapefile. We cannot publish proprietary or copyrighted maps (e.g. Google Maps, Mapquest) and the terms of use for your map base layer must be compatible with our CC-BY 4.0 license. 

Additional Editor Comments (if provided):

Reviewers' comments:

Reviewer's Responses to Questions

**Comments to the Author**

1. Does this manuscript meet PLOS Global Public Health’s publication criteria ? Is the manuscript technically sound, and do the data support the conclusions? The manuscript must describe methodologically and ethically rigorous research with conclusions that are appropriately drawn based on the data presented.

Reviewer #1: Yes

Reviewer #2: Partly

2. Has the statistical analysis been performed appropriately and rigorously?

Reviewer #1: No

Reviewer #2: No

3. Have the authors made all data underlying the findings in their manuscript fully available (please refer to the Data Availability Statement at the start of the manuscript PDF file)?

Reviewer #1: Yes

Reviewer #2: No

4. Is the manuscript presented in an intelligible fashion and written in standard English?

Reviewer #1: Yes

Reviewer #2: Yes

5. Review Comments to the Author

Reviewer #1: Introduction: Introduction has touched multiple angles. Although: The low contraceptive prevalence rate for Africa and Sub-Saharan Africa need to be clarified.(i.e. in Africa, the CPR remains much lower, particularly in Sub-Saharan Africa, where only about 25% of women aged 15-49 use modern contraceptive methods. What is the cut-off point?).

Methods:The methods used, including the study design, area, sample, sampling technique, instrument, and analysis, were valid and reliable, which led to the final results. BUT:

1. Study design and the reason why you selected should be well described.

2. Sampling technique; It is not clear to me how you employ the multistage clustering sampling until you got your sample size.(i.e. describe it step by step)

3. Study instrument is not clear to me.(i.e. Is it adopted, tested and modified to suit your context?)

4. Data collection technique is missing

5. Data analysis should be well described( i.e. how did you summarised numerical and categorical variables, how did you run logistic regression?)

Results: The heading of the figures and tables should be bolded and written below the figure or above the tables. Table 1 and Figure 4's headings are not bolded.

Reviewer #2: Peer Review A Comparative Analysis of Factors Associated with Modern Contraceptive Use among Youth in Northern and Southern Nigeria: A Cross-sectional Population-based Survey (2011-2021)

1. Main Claims and Significance

Main Claims: The paper claims that there are significant regional disparities in modern contraceptive use (MCU) among youth in Northern and Southern Nigeria, driven by factors such as education, wealth, marital status, and religion. It also highlights the need for region-specific interventions to address these disparities.

Significance: These claims are highly significant for the discipline of public health, particularly in the context of reproductive health and family planning. The findings provide critical insights into the barriers to contraceptive use in Nigeria, a country with high maternal mortality and low contraceptive prevalence rates. The study’s emphasis on regional disparities is particularly important for designing targeted interventions.

2. Introduction

Contextualization: The authors do a good job of placing their claims in the context of previous literature, citing studies from Nigeria and other Sub-Saharan African countries. They reference key findings on factors influencing MCU, such as education, wealth, and religion, and compare their results with those of other studies.

Treatment of Literature: The authors generally treat the literature fairly, but there are areas where a more critical engagement would be beneficial. For example, while they cite studies that align with their findings, they could better address discrepancies (e.g., why wealth was not a significant factor in the South, contrary to other studies). A deeper discussion of conflicting findings would strengthen the paper.

The introduction could better emphasize the unique contribution of this study. While it mentions regional disparities, it could explicitly state how this study fills gaps left by previous research (e.g., long-term trends, comprehensive regional analysis). While the introduction provides a good overview of global and regional trends, it could benefit from more specific data on Nigeria's youth population and reproductive health challenges. For example, include more recent statistics on adolescent pregnancy rates or maternal mortality in Nigeria.

3. Data and Analyses

Data Source: The description of the MICS dataset is clear, but it would be helpful to provide more details on the sample size and representativeness of the data. For example, how were the 22,229 participants from the North and 12,951 from the South selected? Were there any exclusions or adjustments made to the data? Describe how you managed missing data for each variable and data quality control measures.

Variables: The independent variables are well-defined, but the dependent variable (MCU) could be further clarified. For example, what specific modern contraceptive methods were included in the analysis?

4. Results

Presentation: The results are well-presented.

Trend Analysis: The trend analysis is insightful, but it could benefit from a more detailed discussion of potential reasons for the fluctuations in MCU over time. For example, were there any policy changes or public health campaigns during this period that could explain the trends?

5. Discussion

Begin with a key findings summary and organize by themes rather than results order. Add a separate section for implications and include future research directions.

Comparative Analysis: The discussion effectively compares the findings with previous studies, but it could be more critical. For example, why do some studies (e.g., Oluwatobi, 2021) find wealth to be a significant factor in both regions, while this study does not? Are there methodological differences or contextual factors that could explain these discrepancies?

Regional Differences: The discussion could delve deeper into the cultural and religious factors influencing MCU in Northern Nigeria. For example, how do traditional beliefs and gender norms affect contraceptive use in this region?

Policy Implications: While the study provides general recommendations, it could offer more specific policy actions. For example, what types of youth-friendly reproductive health services are needed in Northern Nigeria? How can education and awareness campaigns be tailored to address cultural and religious barriers?

6. Limitations

Missing Variables: The study mentions that some important variables (e.g., cultural beliefs, and access to health services) were not captured in the MICS dataset. It would be helpful to discuss how these missing variables might affect the results and suggest ways to address this in future research.

7. Conclusion

Summary of Findings: The conclusion effectively summarizes the key findings, but it could be more concise. Consider reducing the repetition of statistical details and focusing on the broader implications of the study. Add specific recommendations based on findings.

Call to Action: The conclusion could end with a stronger call to action for policymakers and public health practitioners. For example, emphasize the urgency of addressing regional disparities in MCU and the potential impact on maternal and child health outcomes.

8. References

Formatting: Some references have inconsistent capitalization, punctuation, and use of italics.

Accessibility: Some references (e.g., reference 20) have broken or overly long URLs. Consider using shortened, stable URLs or DOIs where available.

Some references are outdated (e.g., 3, 12).

6. PLOS authors have the option to publish the peer review history of their article (what does this mean? ). If published, this will include your full peer review and any attached files.

**Do you want your identity to be public for this peer review?** For information about this choice, including consent withdrawal, please see our Privacy Policy .

Reviewer #1: No

Reviewer #2: **Yes: ** Anebo Getachew

---

## [Decision Letter · Decision Letter 1]

19 Mar 2025

PGPH-D-24-02207R1

A Comparative Analysis of Factors Associated with Modern Contraceptive Use among Youth in Northern and Southern Nigeria: A Cross-sectional Population-based Survey (2011-2021)

Dear Dr. Sanni,

Thank you for submitting your manuscript to PLOS Global Public Health. After careful consideration, we feel that it has merit but does not fully meet PLOS Global Public Health’s publication criteria as it currently stands. Therefore, we invite you to submit a revised version of the manuscript that addresses the points raised during the review process.

We look forward to receiving your revised manuscript.

Kind regards,

Miquel Vall-llosera Camps

Staff Editor

Journal Requirements:

Reviewers' comments:

Reviewer's Responses to Questions

**Comments to the Author**

1. If the authors have adequately addressed your comments raised in a previous round of review and you feel that this manuscript is now acceptable for publication, you may indicate that here to bypass the “Comments to the Author” section, enter your conflict of interest statement in the “Confidential to Editor” section, and submit your "Accept" recommendation.

Reviewer #1: (No Response)

Reviewer #2: All comments have been addressed

2. Does this manuscript meet PLOS Global Public Health’s publication criteria ? Is the manuscript technically sound, and do the data support the conclusions? The manuscript must describe methodologically and ethically rigorous research with conclusions that are appropriately drawn based on the data presented.

Reviewer #1: Yes

Reviewer #2: Yes

3. Has the statistical analysis been performed appropriately and rigorously?

Reviewer #1: Yes

Reviewer #2: Yes

4. Have the authors made all data underlying the findings in their manuscript fully available (please refer to the Data Availability Statement at the start of the manuscript PDF file)?

Reviewer #1: Yes

Reviewer #2: Yes

5. Is the manuscript presented in an intelligible fashion and written in standard English?

Reviewer #1: Yes

Reviewer #2: Yes

6. Review Comments to the Author

Reviewer #1: 1. Introduction: Introduction has touched multiple angles. Although: The low contraceptive prevalence rate for Africa and Sub-Saharan Africa need to be clarified.(i.e. in Africa, the CPR remains much lower, particularly in Sub-Saharan Africa, where only about 25% of women aged 15-49 use modern contraceptive methods. What is the cut-off point?). Please address this comment. The reader needs to know that when you say CPR is lower, there is a point of reference.

2. Methods: The methods used, including the study design, area, sample, sampling technique, instrument, and analysis, were valid and reliable, which led to the final results. All comments were addressed well, except for comment number 02. How was the sample size determined? Was a formula used? Please provide it.

3. Results: Comment was addressed well.

Reviewer #2: The author has made efforts to address the feedback provided during the peer review process.

7. PLOS authors have the option to publish the peer review history of their article (what does this mean? ). If published, this will include your full peer review and any attached files.

**Do you want your identity to be public for this peer review?** For information about this choice, including consent withdrawal, please see our Privacy Policy .

Reviewer #1: No

Reviewer #2: **Yes: ** Anebo Getachew

---

## [Decision Letter · Decision Letter 2]

5 May 2025

A Comparative Analysis of Factors Associated with Modern Contraceptive Use among Youth in Northern and Southern Nigeria: A Cross-sectional Population-based Survey (2011-2021)

PGPH-D-24-02207R2

Dear Dr Sanni,

We are pleased to inform you that your manuscript 'A Comparative Analysis of Factors Associated with Modern Contraceptive Use among Youth in Northern and Southern Nigeria: A Cross-sectional Population-based Survey (2011-2021)' has been provisionally accepted for publication in PLOS Global Public Health.

Best regards,

Julia Robinson

Executive Editor

Reviewer Comments (if any, and for reference):

Reviewer's Responses to Questions

**Comments to the Author**

1. If the authors have adequately addressed your comments raised in a previous round of review and you feel that this manuscript is now acceptable for publication, you may indicate that here to bypass the “Comments to the Author” section, enter your conflict of interest statement in the “Confidential to Editor” section, and submit your "Accept" recommendation.

Reviewer #2: All comments have been addressed

2. Does this manuscript meet PLOS Global Public Health’s publication criteria ? Is the manuscript technically sound, and do the data support the conclusions? The manuscript must describe methodologically and ethically rigorous research with conclusions that are appropriately drawn based on the data presented.

Reviewer #2: Partly

3. Has the statistical analysis been performed appropriately and rigorously?

Reviewer #2: Yes

4. Have the authors made all data underlying the findings in their manuscript fully available (please refer to the Data Availability Statement at the start of the manuscript PDF file)?

Reviewer #2: No

5. Is the manuscript presented in an intelligible fashion and written in standard English?

Reviewer #2: Yes

6. Review Comments to the Author

Reviewer #2: All comments have been addressed

7. PLOS authors have the option to publish the peer review history of their article (what does this mean? ). If published, this will include your full peer review and any attached files.

**Do you want your identity to be public for this peer review?** For information about this choice, including consent withdrawal, please see our Privacy Policy .

Reviewer #2: **Yes: ** Anebo Getachew
